# A single dose of ChAdOx1 Chik vaccine induces neutralizing antibodies against four chikungunya virus lineages in a phase 1 clinical trial

Pedro M. Folegatti [1,6], Kate Harrison [1,6], Lorena Preciado-Llanes [1], Fernando Ramos Lopez [1], Mustapha Bittaye[1], Young Chan Kim [1], Amy Flaxman [1], Duncan Bellamy [1], Rebecca Makinson[1], Jonathan Sheridan [1], Sasha R. Azar [2], Rafael Kroon Campos [3], Mark Tilley[1], Nguyen Tran[1], Daniel Jenkin[1], Ian Poulton [1], Alison Lawrie[1], Rachel Roberts[1], Eleanor Berrie[4], Shannan L. Rossi[2], Adrian Hill [1], Katie J. Ewer [1] & Arturo Reyes-Sandoval [1,5✉]

Chikungunya virus (CHIKV) is a reemerging mosquito-borne virus that causes swift outbreaks. Major concerns are the persistent and disabling polyarthralgia in infected individuals. Here we present the results from a first-in-human trial of the candidate simian adenovirus vectored vaccine ChAdOx1 Chik, expressing the CHIKV full-length structural polyprotein (Capsid, E3, E2, 6k and E1). 24 adult healthy volunteers aged 18–50 years, were recruited in a dose escalation, open-label, nonrandomized and uncontrolled phase 1 trial (registry NCT03590392). Participants received a single intramuscular injection of ChAdOx1 Chik at one of the three preestablished dosages and were followed-up for 6 months. The primary objective was to assess safety and tolerability of ChAdOx1 Chik. The secondary objective was to assess the humoral and cellular immunogenicity. ChAdOx1 Chik was safe at all doses tested with no serious adverse reactions reported. The vast majority of solicited adverse events were mild or moderate, and self-limiting in nature. A single dose induced IgG and T-cell responses against the CHIKV structural antigens. Broadly neutralizing antibodies against the four CHIKV lineages were found in all participants and as early as 2 weeks after vaccination. In summary, ChAdOx1 Chik showed excellent safety, tolerability and 100% PRNT$_{50}$ seroconversion after a single dose.

[1] The Jenner Institute, University of Oxford, Oxford, United Kingdom. [2] Department of Pathology, University of Texas Medical Branch, Galveston, Texas, United States of America. [3] Department of Microbiology and Immunology, University of Texas Medical Branch, Galveston, Texas, United States of America. [4] Clinical Bio-manufacturing Facility, University of Oxford, Oxford, United Kingdom. [5] Instituto Politécnico Nacional, IPN. Av. Luis Enrique Erro s/n. Unidad Adolfo López Mateos, Zacatenco, Mexico City, Mexico. [6] These authors contributed equally: Pedro M. Folegatti, Kate Harrison. ✉email: arturo.reyes@ndm.ox.ac.uk

Since its emergence in Tanzania in 1952[1], and subsequent reemergence in a series of outbreaks in Kenya, the Indian Ocean (2004–2006)[2], and the Americas (2013–2017)[3], Chikungunya virus (CHIKV) has become a major international health concern, with both acute and long-term impacts on public health. CHIKV has been identified in over 100 countries across Africa, Asia, Europe, and the Americas[4].

CHIKV is an RNA alphavirus of the *Togaviridae* family that is transmitted to humans in urban settings by *Aedes aegypti* and *Aedes albopictus* mosquitoes. Both mosquito species have dispersed to all continents, with *Ae. aegypti* present mainly in tropical and subtropical regions and *Ae. albopictus* expanding through temperate regions[5]. Their rapid global expansion accounts for the possibility of an even greater burden of chikungunya fever (CHIKF) beyond tropical regions.

Swift CHIKV outbreaks have recently taken place in Europe, where the East–Central–South African (ECSA) CHIKV lineage has been transmitted by the local *Ae. albopictus* vector[6]. Examples of outbreaks occur in either low- and middle-income (LMIC) or high-income (HIC) countries. For instance, one of the largest recorded outbreaks occurred during 2004–2007 in islands of the Indian Ocean and India[2]. During this outbreak, 5202 new CHIKV cases were reported in one month, between February and March 2005. Nevertheless, seroepidemiology studies indicated that nearly 215,000 people were actually infected within one month, corresponding to 63% of the total Grande Comore Island population, leaving 79% of the cases hospitalized or staying at home[7]. Nine months later, in December 2005, the outbreak had extended to the neighboring region of Reunion Island, resulting in approximately 255,000 cases or 33% of the total population and an estimate of 225 deaths constituting a case-fatality rate for CHIKF of 1/1000 cases[8]. In Italy, an outbreak occurred in 2007, affecting 205 individuals in only two months[9]. Autochthonous cases of CHIKF have also been recorded in France in 2010, 2014, and 2017[10–12], and have spread across local populations in 1–3 months after the identification of the index case. This demonstrates the need for effective actions to control outbreaks, and highlights the impact that CHIKV preventive vaccines would have if they are able to induce effective immunity rapidly after a single vaccine dose.

CHIKV infections result in a wide spectrum of clinical presentations, spanning from asymptomatic to chronic, severe, and even disabling arthritis[13]. Infections are of major concern and have a significant economic impact. Studies have estimated 151,031 CHIKV-related chronic inflammatory rheumatism DALYs (disability adjusted life years) after the 2014 outbreak in the Americas, roughly twice as many as the 69,000 dengue DALYs calculated in 2004 for the same region[14].

We have developed a replication-deficient simian adenoviral vector from chimpanzee origin expressing the entire structural cassette polyprotein of CHIKV. ChAdOx1 Chik is a chimpanzee adenoviral vector vaccine expressing the CHIKV structural proteins: Capsid, E3, E2, 6 K, and E1. We have previously shown, by transmission electron microscopy, that expression of the CHIKV structural cassette in mammalian cells leads to the formation of virus-like particles (VLPs) that resemble wild-type CHIKV particles[15]. This suggests that vaccination with ChAdOx1 Chik can induce the formation of CHIKV VLPs, which mimic the tridimensional antigen structure of CHIKV particles released during CHIKV infections.

In preclinical mouse models, high levels of neutralizing antibodies have been induced upon a single, unadjuvanted ChAdOx1 Chik dose[15,16], eliciting complete protection against a lethal CHIKV challenge[16]. ChAdOx1-vectored vaccines are currently in various stages of clinical development and have been assessed in more than 18,000 volunteers across 18 clinical trials spanning 10

diseases, including Zika (NCT04015648), Chikungunya (NCT03590392), MERS (NCT04170829, NCT04170829), and COVID-19 (NCT04324606, NCT04400838, NCT04444674, and ISRCTN89951424). A consistent safe and immunogenic profile has been observed following vaccination with these ChAdOx1-vectored vaccines. Here we report safety and immunogenicity data from a first-in-human trial of the ChAdOx1 Chik candidate CHIKV vaccine.

## Results

**Study population.** Between 18 July 2018 and 18 October 2019, 24 healthy adult subjects received a single dose of ChAdOx1 Chik at $5 \times 10^9$, $2.5 \times 10^{10}$ or $5 \times 10^{10}$ vp (Fig. 1). Baseline characteristics are summarized in Table 1.

**Vaccine safety.** ChAdOx1 Chik was safe at doses up to $5 \times 10^{10}$ vp with no serious adverse reactions reported. A total of 112 local and systemic solicited adverse events (AEs) were reported. The vast majority of solicited AEs were mild (79/112; 70.54%, 95%CI 61.53–78.18) or moderate (27/112; 24.11%, 95% CI 17.13–32.8) and self-limiting in nature. All solicited AEs were completely resolved within seven days and 94.64% of them had their onset within the first 72 h post vaccination (51.79% at D0, 39.29% at D1, and 3.57% at D2). Injection site pain was the most common local AE, reported by 79.17% of participants and was predominantly mild in severity. Fatigue was the most common systemic AE followed by headache, myalgia and feverishness. The frequencies of local and systemic solicited AEs reported during the first seven days are summarized in Table 2. Median duration of solicited AEs is summarized in Table S1. Only one serious adverse event was reported but was deemed not related with ChAdOx1 Chik.

Four participants reported a short-lived temperature above 37.5 °C within the first 72 h post vaccination (two in the intermediate-dose group and two in the high-dose group). The highest temperature recorded was 38 °C (classified as mild). All febrile episodes resolved within 24 h.

The proportion of moderate and severe AEs was significantly higher in group 3 compared with group 2 (relative risk 3.643, 95% CI 1.817–7.666, $p < 0.001$), but there were no safety concerns, despite the higher reactogenicity.

Unsolicited AEs in the 28 days following vaccination considered possibly, probably or definitively related with ChAdOx1 Chik were predominantly mild in nature and resolved within the follow-up period (Table S3). Unsolicited AEs of note include shivering/chills (one severe at D0 and D1, resolved by D2; one moderate at D0, resolved by D1 and one mild at D1, resolved by D2; all in Group 3); insomnia (one severe at D2, resolved by D3—Group 3) and Lower Back Pain (one severe at D0, resolved by D2—Group 2) laboratory AEs considered at least possibly related with the study intervention were self-limiting and predominantly mild in severity (Table S3).

**Humoral immunogenicity.** Neutralizing antibody titers by $PRNT_{50}$ were blindly measured from all 24 participants. All doses were highly immunogenic upon a single immunization, reaching a 100% seroconversion rate at 14 days against representative isolates from three CHIKV lineages: Indian Ocean Lineage (IOL), West African Lineage (WAf), and Asian Lineage (As). $PRNT_{50}$ titers to La Réunion (IOL), 37997 (WAf), and SV-0444 (As), were significantly increased from baseline and were maintained throughout the 182-day follow-up period. $PRNT_{50}$ to YO111213, from the Asian–American Lineage (AsAm), demonstrated a 100% seroconversion rate on day 28 but slightly lower seroconversion rates on days 14, 56 and 182 (91.6%, 91.6% and

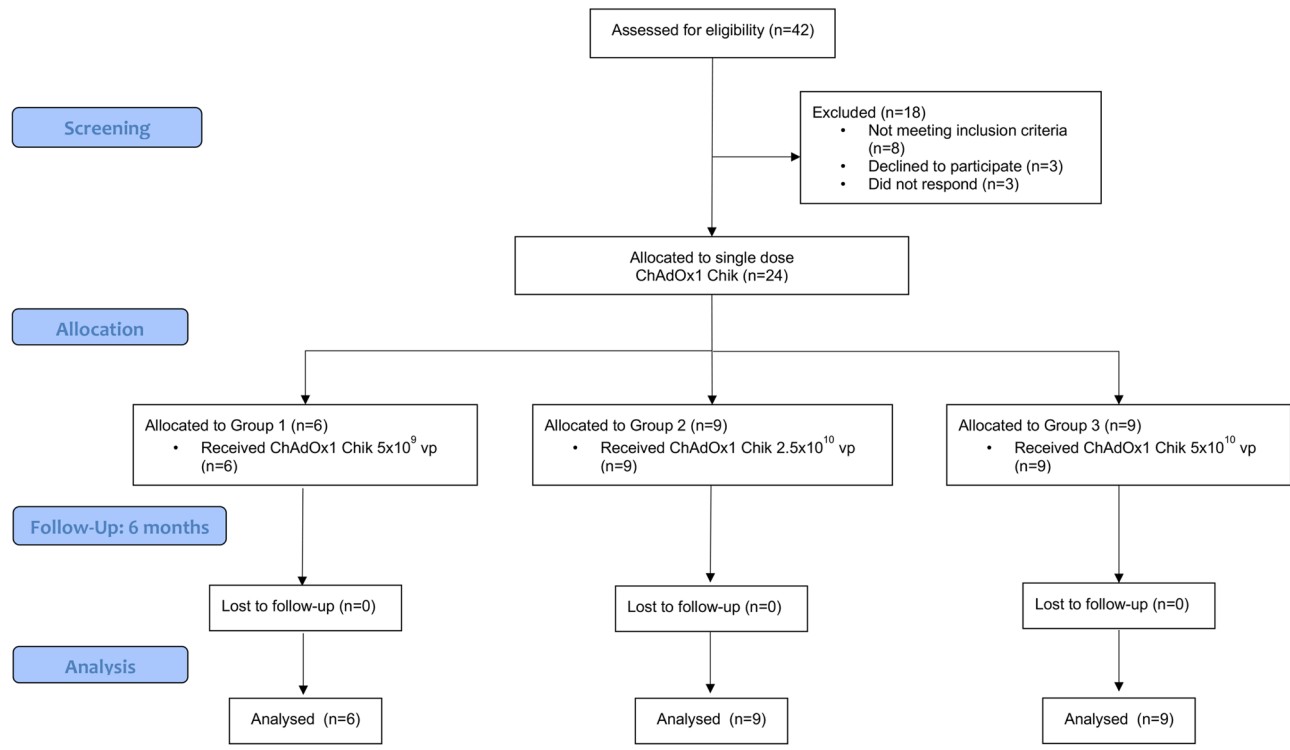

**Fig. 1 Trial profile.** Study profile showing the allocation of participants to three dosage groups: Group 1, low dose at $5 \times 10^9$ vp, Group 2 intermediate dose at $2.5 \times 10^{10}$ vp, and Group 3 high dose at $5 \times 10^{10}$ vp. None of the 24 recruited participants were lost in follow-up.

**Table 1 Summary of participants' baseline characteristics.**

| Variable | Group 1 Low dose ($n = 6$) | Group 2 Intermediate dose ($n = 9$) | Group 3 High dose ($n = 9$) | All Groups ($n = 24$) |
|---|---|---|---|---|
| *Age* | | | | |
| Median | 43 | 26 | 24 | 29 |
| Range | 20–45 | 18–41 | 21–43 | 18–45 |
| *Sex* | | | | |
| Male – *n* (%) | 2 (33·33) | 2 (22·22) | 2 (22·22) | 6 (25) |
| Female – *n* (%) | 4 (66·67) | 7 (77·78) | 7 (77·78) | 18 (75) |
| *Ethnicity* | | | | |
| White – *n* (%) | 6 (100) | 5 (55·56) | 8 (88·89) | 19 (79·17) |
| Mixed[a] – *n* (%) | – | 1 (11·11) | – | 1 (4·17) |
| Latin American – *n* (%) | – | 3 (33·33) | 1 (11·11) | 4 (16·67) |

aMixed: White and Asian.

83.3%, respectively). By day 14, geometric mean titers (GMT) between 40 and 226.3 were measured across the four lineages. GMT peaked at day 28 for IOL (285.5, 95% CI 161.2–504.3), WAf (369.7, 95% CI 217.2–629.3) and AsAm (71.3, 95% CI 49.6–102.4); whereas titers peaked at day 56 for As (75.3, 95% CI 48.0–118.3) (Table 3 and Fig. 2a). An analysis per dose group (Fig. 2b) indicated that all doses are effective at inducing broadly neutralizing antibodies against all CHIKV isolates tested, with PRNT$_{50}$ GMT significantly higher than baseline at almost every time point. The best neutralization was observed against IOL and WAf, with maximum PRNT$_{50}$ values of 1280, 1280 and 2560 at low, intermediate and high vaccine dosages, respectively. In comparison, maximum PRNT$_{50}$ GMT for each dose group were 160, 320, and 1280 for As; 320, 640, and 320 for AsAm (Fig. S2).

Broad cross-neutralizing and protective IgG antibodies, which recognize epitopes on the CHIKV E2 protein have been found in convalescent individuals and in animals, shortly after infection[17–21]. A single ChAdOx1 Chik dose induced high-

antibody titers against CHIKV E2 protein. Geometric mean ELISA units at the peak response were 80.99 (95% CI 38.65–169.7); 205.90 (95% CI 92.66–457.6); 169.70 (95% CI 71.94–400.3) for the low, intermediate, and high dose, respectively (Table 4). The levels of anti-E2 antibodies showed a steady increase over time, reaching maximum seroconversion on day 182: 66.66% (4/6) for the low vaccine dose, 100% (9/9) for the intermediate dose and 77.77% (7/9) for the high dose (Table 4). Compared with day 0, anti-E2 IgG antibody levels started to increase by day 14 ($P = 0.089$), were significantly higher at day 28 ($P = 0.0003$), and reached maximum levels between day 56 and 182 ($P = < 0.0001$) following vaccination (Fig. 3a). Antibodies reached significantly higher levels than baseline as early as day 14 for the high-dose group, day 28 for the intermediate-dose group, and day 56 for the low-dose group (Fig. 3b). It was observed that the calculated cut-off threshold (mean on day $0 + 3$ SDEV), was influenced by four participants with a relatively high ELISA background at baseline. Therefore, we decided to further validate

**Table 2 Number of participants reporting local and systemic solicited AEs.**

| | ChAdOx1 Chik $5 \times 10^9$ vp ($n = 6$) | | | | ChAdOx1 Chik $2.5 \times 10^{10}$ vp ($n = 9$) | | | | ChAdOx1 Chik $5 \times 10^{10}$ vp ($n = 9$) | | | |
|---|---|---|---|---|---|---|---|---|---|---|---|---|
| | Any | Mild | Moderate | Severe | Any | Mild | Moderate | Severe | Any | Mild | Moderate | Severe |
| Any symptom | 4 (67%) | 4 (67%) | 0 | 0 | 9 (100%) | 6 (67%) | 3 (33%) | 0 | 9 (100%) | 4 (44%) | 3 (33%) | 2 (22%) |
| Any local symptom | 3 (50%) | 3 (50%) | 0 | 0 | 9 (100%) | 8 (89%) | 1 (11%) | 0 | 7 (78%) | 5 (56%) | 2 (22%) | 0 |
| Pain | 3 (50%) | 3 (50%) | 0 | 0 | 9 (100%) | 8 (89%) | 1 (11%) | 0 | 7 (78%) | 5 (56%) | 2 (22%) | 0 |
| Pruritus | 0 | 0 | 0 | 0 | 0 | 0 | 0 | 0 | 0 | 0 | 0 | 0 |
| Warmth | 0 | 0 | 0 | 0 | 4 (44%) | 4 (44%) | 0 | 0 | 3 (33%) | 3 (33%) | 0 | 0 |
| Swelling | 0 | 0 | 0 | 0 | 0 | 0 | 0 | 0 | 0 | 0 | 0 | 0 |
| Erythema | 0 | 0 | 0 | 0 | 0 | 0 | 0 | 0 | 1 (11%) | 1 (11%) | 0 | 0 |
| Any systemic symptom | 2 (33%) | 2 (33%) | 0 | 0 | 9 (100%) | 6 (67%) | 3 (33%) | 0 | 9 (100%) | 4 (44%) | 3 (33%) | 2 (22%) |
| Fever | 0 | 0 | 0 | 0 | 2 (22%) | 2 (22%) | 0 | 0 | 2 (22%) | 2 (22%) | 0 | 0 |
| Feverishness | 1 (17%) | 1 (17%) | 0 | 0 | 6 (67%) | 3 (33%) | 3 (33%) | 0 | 7 (78%) | 4 (44%) | 2 (22%) | 1 (11%) |
| Arthralgia | 0 | 0 | 0 | 0 | 3 (33%) | 3 (33%) | 0 | 0 | 4 (44%) | 0 | 3 (33%) | 1 (11%) |
| Myalgia | 1 (17%) | 1 (17%) | 0 | 0 | 7 (78%) | 6 (67%) | 1 (11%) | 0 | 6 (67%) | 2 (22%) | 3 (33%) | 1 (11%) |
| Headache | 1 (17%) | 1 (17%) | 0 | 0 | 7 (78%) | 6 (67%) | 1 (11%) | 0 | 6 (67%) | 3 (33%) | 1 (11%) | 2 (22%) |
| Fatigue | 2 (33%) | 2 (33%) | 0 | 0 | 5 (56%) | 5 (56%) | 0 | 0 | 8 (89%) | 4 (44%) | 4 (44%) | 0 |
| Nausea | 0 | 0 | 0 | 0 | 2 (22%) | 1 (11%) | 1 (11%) | 0 | 3 (33%) | 1 (11%) | 2 (22%) | 0 |
| Malaise | 1 (17%) | 1 (17%) | 0 | 0 | 6 (67%) | 6 (67%) | 0 | 0 | 5 (56%) | 1 (11%) | 3 (33%) | 1 (11%) |

the seronegativity of these individuals with two commercially available ELISA kits. Both antichikungunya virus IgG ELISA kits, from Abcam and Euroimmune, confirmed that none of the participants that had high background in our in-house ELISA were seropositive for CHIKV at baseline (data not shown).

PRNT$_{50}$ against CHIKV IOL showed a significant positive correlation with the measured ELISA units from all dose groups, being the intermediate-dose group the most positively correlated (Spearman's Rho = 0.699 [95%CI 0.5047–0.8269]; $P < 0.0001$) (Fig. 3c). Within the intermediate-dose group, a maximum correlation between PRNT$_{50}$ and ELISA units was reached at day 180 (Spearman's rho = 0.939 [95%CI 0.5047–0.8269]; $P < 0.0003$) (Fig. 3d).

**Cellular immunogenicity.** Cellular immunogenicity was measured from fresh peripheral blood mononuclear cells (PBMC) by an ex vivo IFN-γ ELISpot assay, using pools of peptides spanning the structural CHIKV proteins (Capsid, E3, E2, 6k, and E1) as stimuli. The total responses were quantified at days 0 (baseline), 14, 28, 56, and 182 after vaccination. At baseline, the geometric mean of IFN-γ spot-forming cells (SFC) per million PBMC was 180.1 (IQR 149.9–216.4), across all dosage groups. ChAdOx1 Chik significantly increased the number of IFN-γ SFC, peaking at day 14 post vaccination (1031, IQR 748.9–1420) and remained significantly higher than baseline throughout days 28 (541.1, IQR 411.6–711.13); 56 (398.2, IQR 298.6–530.9); 182 (352.8, IQR 270.1–460.8) (Fig. 4a).

The breadth of the responses against each of the five CHIKV structural antigens was measured by an ex vivo IFN-γ ELISpot assay using pools of overlapping peptides. The responses to all structural proteins, except 6 K, peaked at day 14 after vaccination (Fig. 4b). We observed the largest proportion of responses against E1 and E2, with E2 responses remaining high for a longer period as compared with the other proteins. However, the size of the proteins must be taken into consideration. While E1 and E2 have a similar size and were divided into a similar number of peptides (45 and 42, respectively), the other structural proteins are significantly smaller and had a smaller number of peptides.

Capsid was the third largest protein with 26 peptides, both whereas E3 and 6 K had only 6 peptides each.

Intracellular cytokine staining (ICS) by flow cytometry was carried out at baseline and at day 28 after vaccination. PBMC were stimulated with pools of peptides covering the complete CHIKV structural polypeptide, and analyzed for production of IFN-γ, IL-2, or TNF-α. Analysis by the individual cytokines demonstrated that only IFN-γ producing CD4$^+$ T cells were significantly increased from baseline. An increment on TNF-α$^+$ and IL-2$^+$-producing CD4$^+$ T cells was observed in a proportion of the participants, but did not reach significance. CD8$^+$ T-cell responses did not show significant differences for any of the three cytokines between baseline and day 28 (Fig. 4c).

**Discussion**
We have shown safety and an excellent immunogenicity profile by a novel CHIKV vaccine using the replication-deficient chimpanzee adenoviral vector ChAdOx1 expressing the structural proteins capsid, E3, E2, 6k, and E1 from CHIKV. Our findings demonstrate that the candidate ChAdOx1 Chik vaccine given as a single dose was safe and well tolerated across all doses tested, including $5 \times 10^9$, $2.5 \times 10^{10}$, and $5 \times 10^{10}$ vp. Higher reactogenicity was observed at the highest dose of $5 \times 10^{10}$ vp. No serious adverse reactions to ChAdOx1 Chik occurred. The majority of AEs reported were mild or moderate in severity, and all were self-limiting. We observed transient cases of leukopenia, neutropenia, and lymphopenia in 5, 5 and 1 volunteer, respectively. Most of these were mild and resolved by day 7, one was moderate and resolved by day 28. None of them were severe. The profile of adverse events reported here is similar to that reported for other ChAdOx1 vectored vaccines expressing different antigens[22,23].

CHIKV causes swift outbreaks, affecting large populations and spreading to neighboring regions rapidly. This highlights the requirement for vaccines with the capacity to rapidly stimulate immunity within days after immunization. Nevertheless, most vaccines tested in clinical trials use prime-boost regimens, requiring several weeks to induce immune responses.

**Table 3 PRNT$_{50}$ against CHIKV lineages.**

PRNT$_{50}$

| | Indian Ocean (La Reunion) | | West African (37997) | | Asian (SV0A44-95) | | Asian/American (YO-111213) | |
| --- | --- | --- | --- | --- | --- | --- | --- | --- |
| | % of seroconversion | Geometric mean titers (95% CI) | % of seroconversion | Geometric mean titers (95% CI) | % of seroconversion | Geometric mean titers (95% CI) | % of seroconversion | Geometric mean titers (95% CI) |
| Day 0 | 0% | 5 (5.0–5.0) | 0% | 5 (5.0–5.0) | 0% | 5 (5.0–5.0) | 0% | 5 (5.0–5.0) |
| Day 14 | 100% | 164.7 (102.3–265.2) | 100% | 226.3 (139.9–365.9) | 100% | 47.6 (29.4–77.1) | 91.6% | 40 (25.5–62.6) |
| Day 28 | 100% | 285.1 (161.2–504.3) | 100% | 369.7 (217.2–629.3) | 100% | 67.3 (46.1–98.2) | 100% | 71.3 (49.6–102.4) |
| Day 56$^a$ | 100% | 222.9 (143.3–346.6) | 100% | 251.4 (148.9–424.6) | 100% | 75.3 (48.0–118.3) | 91.6% | 49.4 (28.6–85.2) |
| Day 182 | 100% | 213.6 (126.4–360.8) | 100% | 229.7 (125.4–420.8) | 100% | 95.1 (56.2–161.1) | 83.3% | 42.4 (21.9–82.2) |

PRNT$_{50}$ = serum dilution required to reduce viral plaques by 50% of the control value. Seroconversion was measured as PRNT$_{50}$ values of 10 or greater.
$^a$One day 56 sample, from the low dose group, was excluded from analysis due to QC failure.

Animal and epidemiological studies have shown that protection from CHIKV disease is associated with the induction of neutralizing antibodies[18,24–26], primarily directed against structural proteins[17–19]. A virus-like particle (VLP) vaccine has reported to induce neutralizing antibodies eight weeks following a homologous prime-boost vaccination[27]. Similarly, a measles-vectored vaccine (MV-CHIK), which also requires prime-boost regimens at 4- and 28-week intervals, demonstrated induction of immune responses after 8 and 32 weeks, respectively[28]. Recently, a live-attenuated vaccine (LAV) CHIKV vaccine based on an ECSA lineage has been reported to use a single dose to induce 100% homologous seroconversion at day 14 after a single administration[29]. Our data demonstrate that the ChAdOx1 Chik platform achieves equal levels of 100% seroconversion by PRNT$_{50}$ in only 14 days after a single administration, with evidence of cross-protective functional antibodies against 4 distinctive CHIKV lineages. To our knowledge, only one additional vaccine candidate in clinical trials has reported broad cross-neutralizing responses against isolates from the 3 CHIKV genotypes (ECSA, Asian, and West African) and the Indian Ocean sublineage[30].

The role of T-cell immunity in clearance of CHIKV is not well understood and controversial. Although activation of CHIKV-specific CD4$^+$ and/or CD8$^+$ T cells has been observed upon vaccination and natural infection[31–35], some publications have rejected that they might have a protective role[18,24,36]. Moreover, the presence of reactive cytokine-producing CD4$^+$ T cells in the joints appear to exacerbate disease and lead to the development of arthritogenic disease[37,38]. IFN responses are likely not pathogenic, whereas TNF-α and Th2 cytokines might be[39,40]. ChAdOx1 induced a CD4$^+$ IFN-γ$^+$ biased cellular response toward CHIKV E1 and E2 proteins, but had no significant effect on CD8$^+$ T cells. CD8$^+$ T cells appear to have no protective role[37] and are of limited value in normal settings; only when antibodies are missing can a protective role be seen[41].

The limitations of this study include the relatively short follow-up period of six months, small sample size and an open-labeled, nonrandomized, uncontrolled study design. Generalization of the study findings is limited, as this is a first-in-human study of healthy volunteers. Further studies should be conducted in older and younger age groups, adults with comorbidities, and in populations considered to be at risk of developing chronic arthritis following CHIKV infection.

In conclusion, ChAdOx1 Chik was safe and well tolerated at all tested doses. A single dose showed compelling evidence of rapid stimulation of cellular responses and induction of high titers of functional antibodies with the capacity to neutralize multiple CHIKV lineages. Since our platform does not require an adjuvant and shows to be immunogenic even at a low dose, ChAdOx1 Chik constitutes an attractive product for manufacturers and an affordable preventive vaccine for low-income countries. The ability to induce robust cellular and humoral immunity upon a single administration portraits ChAdOx1 Chik as a suitable candidate to limit swift outbreaks around the world. The results of this first-in-human clinical trial support clinical development progression into phase 1b and 2 trials in CHIKV-endemic regions, such as those in Latin America, India and Africa.

## Methods

**Trial objectives, participants, and oversight**. This is a first-in-human, dose-escalation, open-label, nonrandomized and uncontrolled clinical study of 24 healthy male and female subjects aged 18–50 years old. The sample size was selected based on other previous phase I trials using the same vector. This sample size was able to detect significant differences in immune responses from baseline while exposing a limited number of people to an investigational medicinal product that was being used for the first time.

The primary objective was to assess the safety and tolerability of ChAdOx1 Chik in healthy volunteers, measured as (a) occurrence of solicited local reactogenicity

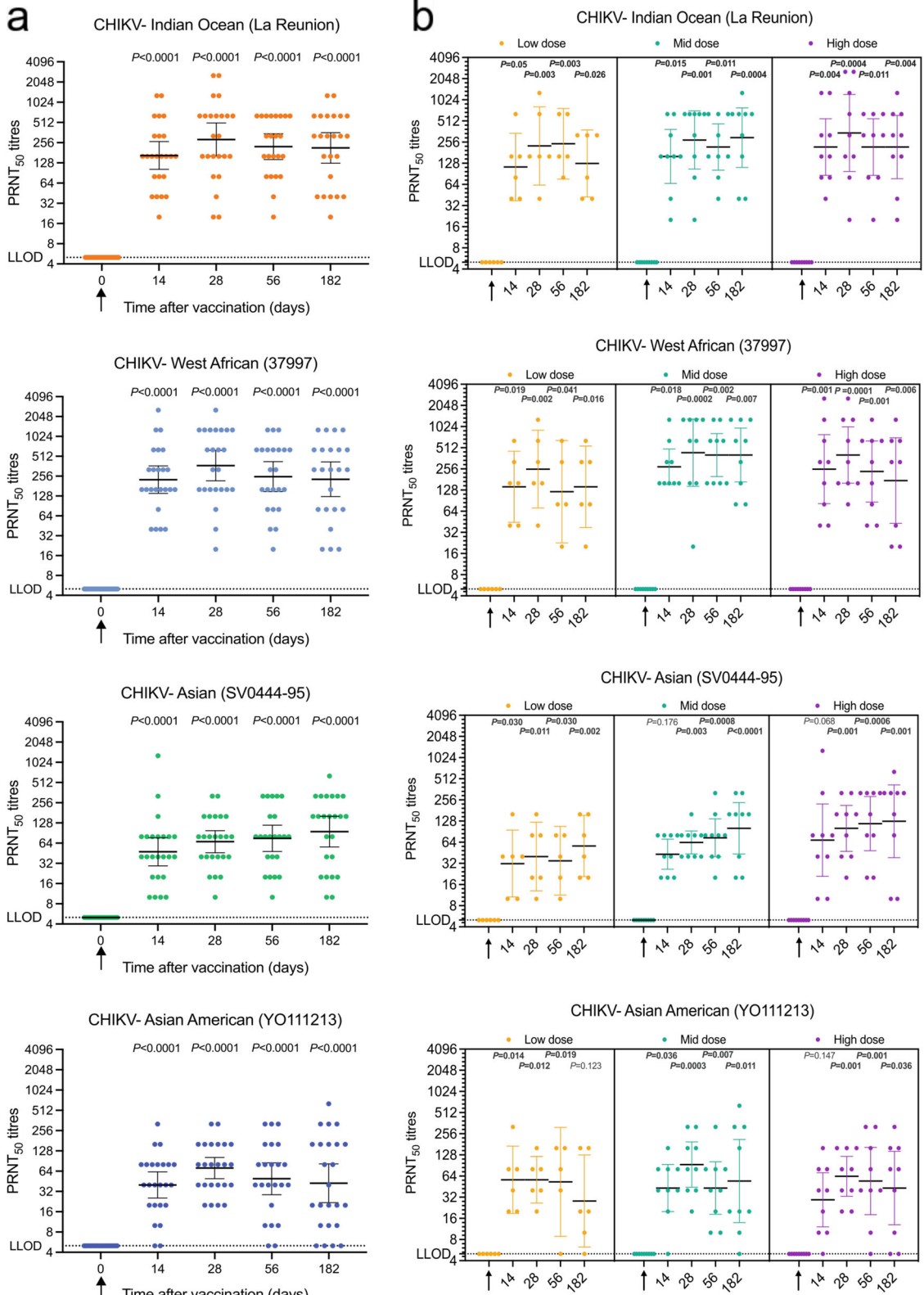

**Fig. 2 PRNT$_{50}$ values over time.** PRNT$_{50}$ reciprocal titers are shown for each participant. Arrows indicate when ChAdOx1 Chik was administered. Lower limit of detection (LLOD) is 5. **a** Overall responses by timepoint in all trial volunteers ($n = 24$). Geometric means and 95% CI; Kruskal–Wallis test with Dunn's correction. **b** Same data as (**a**) but analyzed by dosage groups: low dose was $5 \times 10^9$ vp ($n = 6$), intermediate (mid) dose was $2.5 \times 10^{10}$ vp ($n = 9$), and high dose was $5 \times 10^{10}$ vp ($n = 9$). Geometric means and 95% CI; Kruskal–Wallis test with Dunn's correction. One day 56, the sample from the low-dose group was excluded from analysis due to QC failure.

**Table 4 CHIKV IgG response by standardized ELISA to E2 protein.**

| ELISA | | | | | | |
|---|---|---|---|---|---|---|
| | Low dose (n = 6) | | Intermediate dose (n = 9) | | High dose (n = 9) | |
| | Seropositivity n (%) | Geometric mean titerss (95% CI) | Seropositivity n (%) | Geometric mean titerss (95% CI) | Seropositivity n (%) | Geometric mean titerss (95% CI) |
| Day 0 | 0 (0%) | 4.741 (0.806–27.87) | 0 (0%) | 4.725 (1.392–16.04) | 0 (0%) | 3.009 (1.231–7.354) |
| Day 14 | 2 (33.33%) | 7.475 (1.075–51.95) | 3 (33.33%) | 13.03 (4.078–41.66) | 0 (0%) | 12.14 (6.449–22.84) |
| Day 28 | 1 (16.66%) | 28.83 (19.10–43.52) | 4 (44.44%) | 44.51 (23.90–82.90) | 1 (11.11%) | 29.18 (21.07–40.42) |
| Day 56 | 3 (50%) | 41.67 (21.51–80.74) | 6 (66.66%) | 87.10 (42.56–178.2) | 7 (77.77%) | 70.02 (45.76–107.1) |
| Day 182 | 4 (66.66%) | 80.99 (38.65–169.7) | 9 (100%) | 205.9 (92.66–457.6) | 7 (77.77%) | 169.7 (71.94–400.3) |

Data are geometric mean with 95% CI. Analysis was made using Friedman with Dunn's. Cut-off = average of days 0 + 3 SDEV.

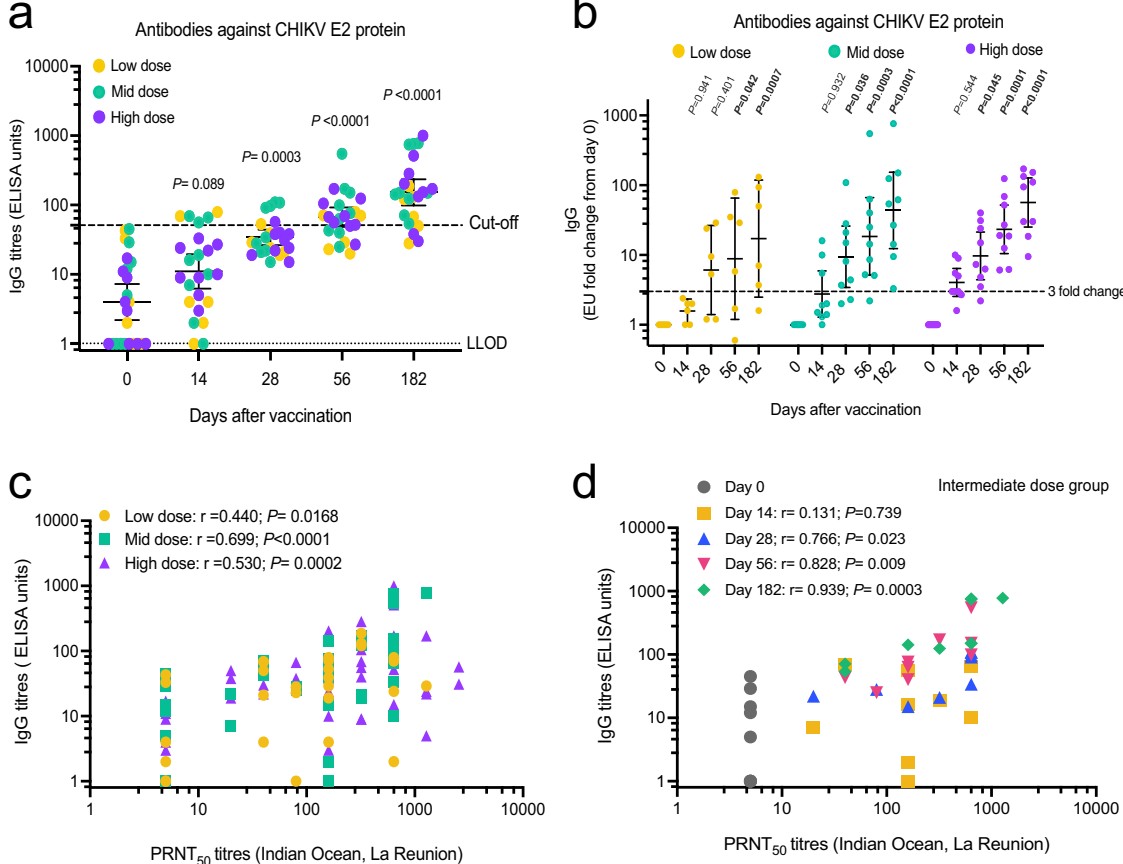

**Fig. 3 ELISA titers over time.** CHIKV IgG response by standardized ELISA to E2 protein in 120 serum samples of trial participants. **a** Individual IgG titers over time (n = 24). The dashed line represents the cut-off value for seropositivity. Geometric means and 95% CI; Friedman test with Dunn's correction. **b** Same data as (**a**) but represented as fold change from baseline (day 0) and analyzed by dosage group: low dose was 5 × 10⁹ vp (n = 6), intermediate (mid) dose was 2.5 × 10¹⁰ vp (n = 9), and high dose was 5 × 10¹⁰ vp (n = 9). Geometric means and 95% CI; Friedman test with Dunn's correction. **c** Correlation of PRNT50 and IgG ELISA by dosage group at 5 time points. Low dose, six participants, n = 29 (on day 56, samples were excluded due to QC failure); intermediate (mid) dose, nine participants (n = 45); high dose, nine participants (n = 45). Spearman correlation, two-tailed. **d** Same data as (**c**) but correlation is only shown for the nine participants vaccinated at the intermediate (mid) dose (n = 9 per timepoint). Spearman correlation, two-tailed.

signs and symptoms for seven days following vaccination; (b) occurrence of solicited systemic reactogenicity signs and symptoms for seven days following vaccination; (c) occurrence of unsolicited adverse events (AEs) for 28 days following vaccination; (d) change from baseline for safety laboratory measures; (e) occurrence of serious adverse events (SAEs) during the whole study duration. The secondary objective was to assess CHIKV structural antigen-specific humoral and cellular immune responses induced by ChAdOx1 Chik as measured by enzyme-linked immunosorbent assay (ELISA), plaque reduction neutralization test (PRNT) and ex vivo interferon-gamma (IFN-γ) enzyme-linked immunospot (ELISpot).

Eligible volunteers were recruited at the Centre for Clinical Vaccinology and Tropical Medicine, Oxford, United Kingdom (CONSORT diagram, Fig. 1). All

participants were healthy adults with negative prevaccination tests for HIV antibodies, hepatitis B surface antigen, and hepatitis C antibodies. A negative urinary pregnancy test was required at screening and immediately before enrollment for all female subjects. Screening for previous CHIKV exposure was conducted on participants with significant travel history to CHIKV-endemic areas, using a commercial ELISA kit (Anti-Chikungunya Virus IgG Human ELISA Kit, Abcam ab177835) and were excluded if positive. Full details of the eligibility criteria are described in the trial protocol provided in the Supplementary Materials.

The corresponding author had full access to all the data in the trial and had final responsibility for the decision to submit the paper for publication. All the trial data were available to all the authors.

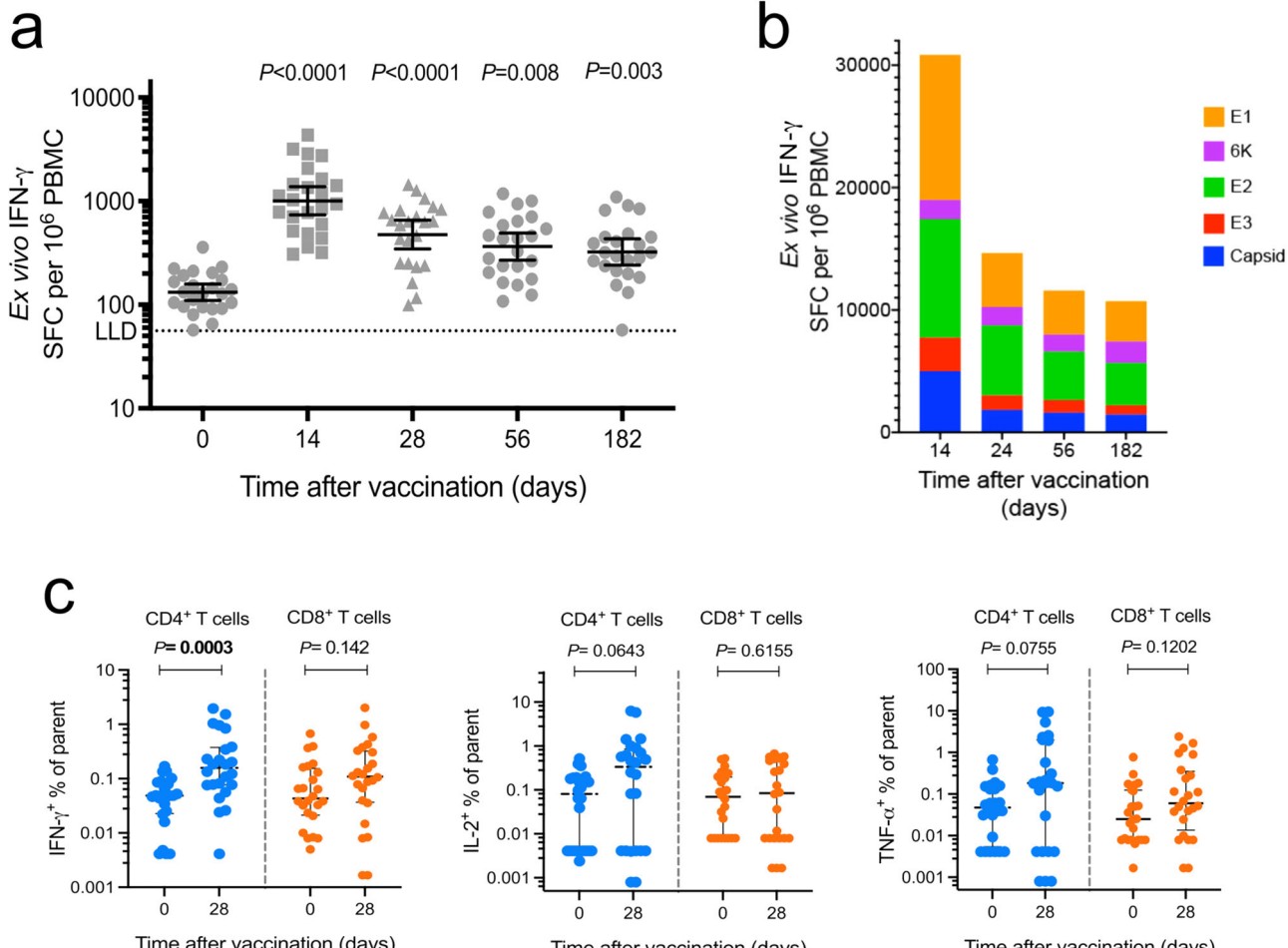

**Fig. 4 T-cell responses over time. a** Ex vivo enzyme-linked immunospot (ELISpot) for IFN-γ to CHIKV structural antigens measured as total responses to CHIKV peptides (sum of 13 pools spanning C, E3, E2, 6 K, and E1). SFC per million PBMC during a 6-month follow-up period (n = 24). Median and IQR; Kruskal–Wallis test with Dunn's correction. **b** Proportion of spots contributed by C, E3, E2, 6 K and E1 over time. **c** Intracellular cytokine staining (ICS) by flow cytometry to assess CD4+ and CD8+ T-cell functionality. Percentage of cytokine-producing CD4+ and CD8+ T cells (n = 24). Median and IQR; Mann–Whitney test, two-tailed.

**Study approvals**. Written informed consent was obtained in all cases, and the trial was conducted in accordance with the principles of the Declaration of Helsinki and Good Clinical Practice (GCP). This study was approved within the United Kingdom by the Medicines and Healthcare Products Regulatory Agency (MHRA reference 21584/0394/001-0001) and the South Central Oxford A Research Ethics Committee (REC reference 18/SC/0004). Vaccine use was authorized by the Genetically Modified Organisms Safety Committee of the Oxford University Hospitals National Health Service Trust (GMSC reference number GM462.18.102). The trial is registered at www.clinicaltrials.gov (identifier: NCT03590392). The first participant was enrolled on 02 October 2018 and the last participant was enrolled on 01 April 2019.

**Trial procedures**. ChAdOx1 Chik was administered as a single intramuscular injection into the deltoid at a low dose of $5 \times 10^9$ vp (group 1), intermediate dose of $2.5 \times 10^{10}$ vp (group 2), and high dose of $5 \times 10^{10}$ vp (group 3). A staggered-enrollment approach was used for the first 3 participants in each group and interim-safety reviews conducted prior to dose escalation (details provided in study protocol).

Blood samples were drawn and clinical assessments conducted for safety as well as immunology endpoints prior to vaccination at day 0 and subsequently at 2, 7, 14, 28, 56, and 182 days following enrollment. Participants were observed in the clinic for one hour after the vaccination procedure and were asked to record any AEs using electronic diaries during the 28-day follow-up period. Swelling at the injection site was objectively assessed by a member of the study team during the study visits.

Expected and protocol-defined local site reactions (injection-site pain, warmth, redness and pruritus) and systemic symptoms (malaise, myalgia, arthralgia, fatigue, nausea, headache, feverishness, and temperature) were recorded for seven days.

Unsolicited AEs were recorded for 28 days and SAEs were recorded throughout the follow-up period.

The severity of AEs was graded using the following criteria: (a) mild (short-lived or mild symptoms with no limitation to usual activity); (b) moderate (mild-to-moderate limitation in usual activity); (c) severe (considerable limitation in activity, medication, or medical attention required). Unsolicited AEs were reviewed for causality by an independent clinician and events considered possibly, probably, or definitively related with the study vaccine were reported. Laboratory AEs were graded using site-specific toxicity tables that were adapted from the US Food and Drug Administration toxicity grading scale. An independent local safety monitor (LSM) provided safety oversight. The relevant clinical data were recorded in a study database using OpenClinica (Enterprise Edition) v3.13.

**ChAdOx1 Chik Vaccine**. ChAdOx1 Chik uses the replication-deficient adenovirus vector derived from the E1 E3-deficient ChAdY25[42] and is currently a leading vaccine platform against COVID-19[43]. ChAdOx1 Chik was engineered to express the full structural polyprotein genome of CHIKV that includes the Capsid, E3, E2, E1, and the 6 K proteins. The synthetic gene was designed through an analysis of full-length structural polyprotein sequences from multiple CHIKV lineages. Sequences were collected from the NCBI database and aligned using Clustal Omega and a neighbor-joining tree (Juke-Cantor, 100 bootstraps). Intra- and interclade conservation was calculated using a sliding-window approach with a sequence-weighting method to enable equal representation of all lineages and variants. A synthetic gene cassette was produced by GeneArt® (ThermoFisher Scientific), which was subsequently cloned into a pMono plasmid to be driven by a CMV promoter expression[15]. The vaccine was manufactured to current Good Manufacturing Practice (cGMP) by the Clinical Biomanufacturing Facility (University of Oxford, Oxford, UK) in a HEK 293-cell line. The vectored vaccine was purified and

sterile-filtered to generate a clinical lot at a concentration of $1.57 \times 10^{11}$ viral particles per mL.

**ELISA.** Total anti-CHIKV IgG was measured using a standardized in-house indirect ELISA[44]. To this end, 1 µg/ml of CHIKV E2-recombinant protein in phosphate-buffered saline (PBS) was used to coat Nunc-immuno 96-well plates. Plates were incubated at 4 °C for 18 h overnight[15,45]. Coated plates were washed six times with PBS-Tween followed by blocking with casein for 1 h at room temperature (RT). Serum samples were diluted at 1:100 or 1:500 in casein to fit within the linear range of a standard curve prepared as indicated below, and then added to individual wells in triplicates. Plates were incubated at RT for 2 h, washed, as described, and then incubated at RT for 1 h with an alkaline phosphatase-conjugated goat anti-human IgG (gamma-chain specific, Sigma). Plates were developed by adding 4-nitrophenyl phosphate (Sigma) in diethanolamine substrate buffer (Thermo Scientific). A standard curve was prepared from a serum sample of a convalescent individual, following a 2-fold serial dilution starting at 1:100 and generating 10 standard points to which arbitrary ELISA units (EUs) were assigned. The optical density (OD) values of the standard points were fitted to a 4-parameter hyperbolic curve against the arbitrary EUs using the BioTek Gen5 v3.09 software and the parameters estimated from the standard curve were used to convert absorbance values of individual test samples into EU. Each ELISA plate contained the samples in triplicates, an internal positive control at 1:1600 dilution of the standard pool in triplicates, 10 standard points in duplicates, and four blank wells. Absorbance reading at 405 nm was performed using an ELx808 microplate reader (BioTek). The assay cutoff was determined from the analysis of the 24 prevaccinated (Day 0) samples of the trial volunteers. The seropositive cut-off was determined mathematically using the mean plus three standard deviations of the EU values reported for the 24 samples assayed. This value defined the threshold from which detection was feasible. A cut-off value of 51.1 EU was used as the analytical sensitivity of this assay.

Two commercially available ELISA kits were used to validate seronegativity of serum samples that had a higher background in our in-house E2 ELISA at baseline. The antichikungunya virus IgG ELISA kit from Abcam (ab177835) and the antichikungunya virus IgG ELISA kit from Euroimmune (EI 293a-9601G) were both performed according to the manufacturer's instructions.

**Plaque reduction neutralization tests (PRNT).** Induction of serum neutralizing antibodies was evaluated with a plaque reduction neutralization tests (PRNT) on monolayers of Vero cells (Vero ATCC CCL-81) cultured in 12-well plates using standard methods[46,47]. Neutralizing antibody titers were recorded for four CHIKV lineages, including the chikungunya strains CHIKV-LR (Indian Ocean Lineage, IOL), SV-0444 (Asian Lineage), 37997 (West African Lineage), and YO111213 (Asian/American Lineage), all obtained from the World Reference Center for Emerging Viruses and Arboviruses at UTMB[46]. Titers were quantified as the highest serum dilution that inhibited plaque formation in 50% ($PRNT_{50}$). Seroconversion was considered positive in samples with reciprocal titers of $PRNT_{50} \geq 10$[48]. Limits of detection were between 10 and 1280, and any samples without a detectable titer were listed as either 5 or 2560.

**ELISpot.** Cellular immune responses were quantified at the selected timepoints using an ex vivo enzyme-linked immunospot (ELISpot) for IFN-γ[43,44]. PBMC were stimulated with 125 synthetic peptides (20mers overlapping by ten amino acids), divided into 13 peptide pools spanning the entire vaccine insert of the CHIKV structural antigens: Capsid (3 pools = 26 peptides), E3 (1 pool = 6 peptides), E2 (4 pools = 42 peptides), 6k (1 pool = 6 peptides) and E1 (4 pools = 45 peptides). Peptide sequences and pooling are summarized in Supplementary Table S4. Data were analyzed according to a quality-control standard operational procedure. The lower limit of detection for the assay was 4 spot-forming cells (SFCs) for summed responses to the 13 CHIKV structural antigen peptide pools. The following antibodies were used for ELISpot assay: anti-human IFN-γ-capture IgG1 mouse monoclonal Ab (dil 1:100) and anti-human IFN-γ-biotinylated, detection mouse IgG1 monoclonal Ab (dil 1:1000).

**Flow cytometry.** Intracellular cytokine staining for flow cytometry (ICS) was performed to quantify $CD4^+$ and $CD8^+$ T-cell responses to the vaccine[49]. Five peptide pools from the structural CHIKV cassette were used as stimuli (Table S4). The representative gating strategy is shown in Fig S1. For the stimulation and staining, the following antibodies were used: anti-human CD14 eFluor450 (dil 1:100), anti-human CD19 eFluor450 (dil 1:100), anti-human CD3 AF700 (dil 1:50), anti-human CD4 APC (dil 1:25), anti-human CD8a APC eFluor780 (dil 1:10), anti-human IFN-γ FITC (dil 1:250), anti-human TNF-a PE-Cy7 (dil 1:500), anti-human IL-2 PE (dil 1:50), anti-human CD28 (1 µg/mL), and anti-human CD49d (Integrin alpha 4) (1 µg/mL). Samples were run in a LSRFortessa (Becton Dickinson); FACSDiva v 8.02 (BD Biosciences) and FlowJo v10.6.2 (BD Biosciences) were used for data recording and analysis, respectively.

**Statistics.** Safety endpoints are described as frequencies with their respective percentages alongside 95% confidence intervals (CI). The association between the frequency of moderate or severe solicited AEs and group allocation (groups 2 and 3) is reported as relative risk with the respective 95% CI and p value (Fisher's exact test). A Kruskal–Wallis test with Dunn's correction for multiple tests was used to assess the CHIKV ex vivo ELISpot IFN-γ responses, whereas a two-tailed Mann–Whitney test was used for ICS data. ELISA and PRNT data were analyzed by either Kruskal–Wallis test or Friedman test with Dunn's correction for multiple parameters, as appropriate. A $P$ value $< 0.05$ was considered significant. Statistical analysis of safety and immunogenicity data was conducted using GraphPad Prism version 9.1 (GraphPad Software Inc., California, USA).

**Reporting summary.** Further information on research design is available in the Nature Research Reporting Summary linked to this article.

## Data availability
There is a restriction on the availability of the data presented on this paper due to the data being used to feed a patent application and because data will be linked to an ongoing Phase 1b blinded study funded by a different research award. Anonymized participant data may be available upon requests directed to the corresponding author (arturo. reyes@ndm.ox.ac.uk). Proposals will be reviewed and approved by the sponsor (CTRG— https://researchsupport.admin.ox.ac.uk/ctrg#/), principal investigator, and collaborators on the basis of scientific merit. If approved and upon signature of a data-access agreement, data can be shared through a secure online platform. Data sharing may take a period of up to six weeks from receiving the request. All data will be made available for a minimum of five years from the end of the trial. The study protocol is available with this publication as part of the supplementary material.

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

## Acknowledgements

The funders of the study had no role in the study design, data collection, data analysis, data interpretation, or writing of the report. This report is independent research funded by Innovate UK (project Nos. 972212 and 971557). The views expressed in this publication are those of the author(s) and not necessarily those of Innovate UK. PMF received financial support from the Coordenacao de Aperfei-coamento de Pessoal de Nivel Superior, Brazil (finance code 001). We would like to thank the support of Brian Angus (local safety monitor). Yrene Themistocleous, Julia Marshall, and Mehreen Datoo (clinicians). Megan Baker, Celia Mitton, Raquel Lopez Ramon, and Lucy Kingham Page (nurses). Daniel Marshall Searson (data manager). Yara Neves Silva (clinical trial assistant). Natalie Lella and Michelle Fuskova (recruitment coordinators).

## Author contributions

P.M.F., A.H,. and A.R.S. designed the study. P.M.F., F.R.L., D.J., I.P., and R.R. collected study data and oversaw participant visits. A.L. provided regulatory oversight, and M.T., N.T., and R.R. provided project management. Immunogenicity testing was done and interpreted by K.H., M.B., L.P.L., A.F., D.B., R.M., J.S., K.E., and A.R.S. Y.C.K. produced purified protein for ELISA and performed immunopotency assay of the GMP batch. The analysis of samples by PRNT assays was designed, done and interpreted by S.R.A., R.K.C. and S.L.R. Clinical trial data management was done by PMF and IP. Safety data analysis and interpretation were done by P.M.F. P.M.F., K.H,. L.P.L., and A.R.S. wrote the paper. A.H. was the chief investigator. E.B. was responsible for vaccine manufacture. A.R.S. applied and obtained funding to support this project. All authors contributed to the reviewing and editing of the report and approved the final version.

## Competing interests

Pedro M. Folegatti is a consultant to Vaccitech, which is developing adenoviral vectored vaccines. Adrian Hill is a cofounder of and consultant to Vaccitech Ltd and is named as an inventor on a patent covering the design and use of Ch.
