## [Peer Review File · Nature Communications]

Reviewers' Comments:

Reviewer #1:

Remarks to the Author:

Fig. 4d. The assumption that more cytokines made by T cells the better is not shown for CHIKV and is unhelpful in this context. CD4 T cells are arthritogenic (J Immunol. 2013 Jan 1;190(1):259-69) and drive disease. The cytokines made by CD4 T cells that are protective vs those that are arthritogenic are not entirely clear, IFN γ is probably not pathogenic, whereas TNF α may be (Life Sci Alliance. 2019 Feb; 2(1): e201900298). Thus the number (1, 2 or 3 cytokines) is largely unhelpful, with the specific type of cytokine(s) being made much more important; e.g. if green is TNF, low dose might avoid induction of potentially arthritogenic T cells. CD8 T cells also appear to have no protective role (J Immunol. 2013 Jan 1;190(1):259-69) and are thus of limited value in normal settings; only when other immune responses are missing can a protective role be seen (J Immunol. 2015 Jan 15; 194(2): 678-689). These issues should be discussed and the data in Fig. 4 presented in such a way that the % of cells making specific cytokines is revealed.

L48 The number of countries is over 100 see Nat Rev Rheumatol. 2019 Oct;15(10):597-611.

L 152 This hierarchy surely needs to be adjusted for size of the protein. The larger the size the more epitopes are likely. I suspect there would be few significant differences in epitope density. Chi squared tests would be needed before hierarchies can be presented, although epitope density would be interesting to see.

L175-193 should be moved to introduction, not really discussion.

L196 Not sure what heterologous is supposed to mean here, presumably inferring a broad cross neutralising response to multiple lineages. Needs to be rephrased. Also the statement is not true, see NPJ Vaccines. 2020 Jun 2;5(1):44, where response to 4 lineages are shown.

L172 some punctuation missing?

Reviewer #2:

Remarks to the Author:

In the submitted manuscript, Folegatti et al. present the results of a phase I clinical study of ChAdOx1 Chik vaccine candidate that expresses CHIKV structural proteins. The study design and endpoints are similar to other phase I trials, as expected, and the manuscript is well-organized and clearly written. That the vaccine elicits 100% PRNT seroconversion after only one dose is a significant achievement. I have a few comments for clarification:

In the introduction or discussion, it would be helpful to include a description of the CHIKV antigen that is produced and presented following administration of the ChAdOx1 Chik vaccine. While prior in vitro and animal studies have likely addressed this in detail, a reminder that expression of the CHIKV structural proteins results in VLP formation would be helpful for a more broad audience. Similarly, suggest defining the abbreviation DALYs (line 64). In the introduction, the authors list the numerous vaccine candidates that utilize the ChAdOx1 vector strategy- will vaccination with one ChAdOx1 vectored vaccine impact a second vaccination targeting a different pathogen?

The cut-off for the CHIKV E2 ELISA in Figure 3 should be better explained. It looks as though the cut-off was made just above the highest day 0 sample, yet there is significant variability in the day 0 samples. It seems counter-intuitive to have lower ELISA seroconversion compared to PRNT50 seroconversion. Have the authors analyzed the data by normalizing to each individual subject's day 0 titer as a baseline? In panels c and d, would the PRNT50 data be better suited on a log x-axis?

Minor edits:

Line 116- AsAm- the abbreviation has not been introduced.

Line 115- "Slightly lower seroconversion rates of 96.7%, 100% and 83.3% were observed against YO111213 (AsAm) on days 14, 28, and 182."

I'm assuming the authors meant to include data for day 56 and not day 28? A 100% seroconversion rate is not lower than 100%! Also, 96.7% does not match the values in Table 3, which has 91.6%, 91.6% and 83.3 % seroconversion at days 14, 56 and 182, respectively, for the AsAm strain.

Line 122- Should "maximum PRNT50 GMT" be simply "maximum PRNT50"? It is either the maximum PRNT50 value chosen from all of the individual values within the group, or the GMT of the group, but these are exclusive data points. Please explain if otherwise.

Line 137- typo- Withing should be within

Reviewer #1

We thank the reviewer for the careful and considered critique of our initial submission.

- 1) Fig. 4d. The assumption that more cytokines made by T cells the better is not shown for CHIKV and is unhelpful in this context. CD4 T cells are arthritogenic (J Immunol. 2013 Jan 1;190(1):25969) and drive disease. The cytokines made by CD4 T cells that are protective vs those that are arthritogenic are not entirely clear, IFNg is probably not pathogenic, whereas TNFa may be (Life Sci Alliance. 2019 Feb; 2(1): e201900298). Thus the number (1, 2 or 3 cytokines) is largely unhelpful, with the specific type of cytokine(s) being made much more important; e.g. if green is TNF, low dose might avoid induction of potentially arthritogenic T cells. CD8 T cells also appear to have no protective role (J Immunol. 2013 Jan 1;190(1):259-69) and are thus of limited value in normal settings; only when other immune responses are missing can a protective role be seen (J Immunol. 2015 Jan 15; 194(2): 678–689). These issues should be discussed and the data in Fig. 4 presented in such a way that the % of cells making specific cytokines is revealed.** It is true that there is no conclusive evidence to support that polyfunctional memory T cells might be more effective in clearing CHIKV. However, some reports have shown the induction of antigen-specific polyfunctional CD4 and CD8 T cells following natural infection and immunisation with candidate CHIKV vaccines (*PLOS One* 2013 Dec 8(12):e84695, *J Virol* 2014 Mar 88(6); *JCI Insight* 2017 Mar 2(6) e83527). We have included a new paragraph (lines 229-238) where the role of T cells in viral clearance is discussed.

We take the reviewer's comment as a valid point and we acknowledge that it is essential to present the data per individual cytokine. We have removed panels 4c and 4d and replaced them with a new panel that demonstrates the % of CD4 and CD8 T cells producing each of the 3 cytokines (IFN γ , TNF α and IL-2). Such analysis supports a strong IFN γ primary response in CD4 $^+$ T cells. The overall increment on TNF α^+ and IL-2 $^+$ CD4 $^+$ T cells was not significantly different from baseline. CD8 $^+$ T cells did not show significant differences for any of the 3 cytokines. Such finding suggest that our candidate vaccine induces a targeted CD4 $^+$ T-cell memory response that doesn't resemble an arthritogenic profile. In our view, we considered relevant to investigate and report both CD4 and CD8 T cell responses to support our IFN γ responses detected by ELISpot.

Since this a FIH trial, the author's intention was to portrait a general overview of both cellular and humoral responses triggered by ChAdOx1 Chik. It was never our intention to enforce T cell responses as a correlate of protection. Instead, as per many published reports, we consider that neutralising antibodies are the main correlate of protection against CHIKV.

2) **L48 The number of countries is over 100 see Nat Rev Rheumatol. 2019 Oct;15(10):597-611.** We have amended the statement and included the suggested up to date reference.

3) **L152 This hierarchy surely needs to be adjusted for size of the protein. The large the size the more epitopes are likely. I suspect there would be few significant differences in epitope density. Chi squared tests would be needed before hierarchies can be presented, although epitope density would be interesting to see.**

We appreciate the comment and acknowledge that the term 'hierarchy' was used incorrectly. We did not investigate responses per individual peptide but by peptide pools. Therefore, we are unable to provide a detailed immunodominance assessment and/or epitope density.

The statement has now been modified to highlight the temporal responses (as proportions) for each of the structural proteins. A reference on the different protein sizes has also been included.

4) **L175-193 should be moved to introduction, not really discussion.**

We have moved most of these lines to the introduction.

5) **L196 Not sure what heterologous is supposed to mean here, presumably inferring a broad cross neutralising response to multiple lineages. Needs to be rephrased. Also the statement is not true, see NPJ Vaccines. 2020 Jun 2;5(1):44, where response to 4 lineages are shown.**

We have now replaced the term 'heterologous' for 'broad cross-neutralising'.

We thank the reviewer for pointing out a relevant reference. However, we were only referring to candidate vaccines that are already in clinical phase. We have now amended the statement to avoid misunderstanding.

6) **L172 some punctuation missing?**

The sentence has been rephrased to 'Most of these were mild and resolved by day 7, one was moderate and resolved by day 28'.

Reviewer #2

We thank the reviewer for the careful and considered critique of our initial submission.

- 1) *In the introduction or discussion, it would be helpful to include a description of the CHIKV antigen that is produced and presented following administration of the ChAdOx1 Chik vaccine. While prior in vitro and animal studies have likely addressed this in detail, a reminder that expression of the CHIKV structural proteins results in VLP formation would be helpful for a more broad audience.***

As the reviewer points out, ChAdOx1 Chik does induce VLP formation. Evidence of this has been previously published by our group (Camacho et al. *Viruses*. 2019 Apr 11(4):322).

We have now included the following sentences in the introduction: '*ChAdOx1 Chik is a chimpanzee adenoviral vector vaccine expressing the CHIKV structural proteins: Capsid, E3, E2, 6K and E1. We have previously shown, by transmission electron microscopy, that expression of the CHIKV structural cassette in mammalian cells leads to the formation of virus-like particles (VLPs) that resemble wild-type CHIKV particles (Camacho et al. *Viruses*. 2019 Apr 11(4):322). This suggests that vaccination with ChAdOx1 Chik can induce the formation of CHIKV VLPs, which mimic the tridimensional antigen structure of CHIKV particles released during CHIKV infections*'.

- 2) *Similarly, suggest defining the abbreviation DALYs (line 64).***

The definition for this abbreviation has now been included.

- 3) *In the introduction, the authors list the numerous vaccine candidates that utilize the ChAdOx1 vector strategy- will vaccination with one ChAdOx1 vectored vaccine impact a second vaccination targeting a different pathogen?***

Published data on our ChAdOx1 vectored vaccine against SARS-Cov-2 (Vaxzebrina, Astra Zeneca), shows evidence of augmented neutralising antibodies being induced after a booster immunisation with the same ChAdOx1 vector (Folegatti, et.al. *Lancet*. July 2020 396: 467–78; Voysey et.al. *Lancet*. 2021 397:99-111). A large analysis from four trials, concluded that efficacy was higher in those with a longer prime-boost interval (at >12 weeks) than in those with a short interval (<6 weeks) (Voysey et.al. *Lancet*. 2021 Mar 397:881-891). Such findings reflect a vectored-induced immunity which is short lasting and does not cause a detrimental impact on the target antigenic responses.

Other studies using the adenoviral platform for different pathogens have had similar observations and suggest that immune responses can be boosted even when using the same adenoviral vector for prime and boost (Barouch et. al. *Lancet* 2018, 392(10143):232-243; Capone et al. *NPJ Vaccines* 2020, 5:94).

The Jenner Institute is currently investigating the impact of ChAdOx1 vectored vaccines against different pathogens when administered simultaneously or apart. An ongoing Phase 1b double blinded trial assesses the safety and immunogenicity of two different ChAdOx1 vaccines, one for chikungunya virus and another one for Zika virus, when administered alone or in combination ([ClinicalTrials.gov](https://clinicaltrials.gov/ct2/show/study/NCT04440774) NCT04440774). Results from this study will inform the decision on whether poly-disease vaccination with the same vector is possible. Finally, the Jenner Institute is also planning a clinical trial in which recipients of the Vaxzebrina Astra Zeneca vaccine will be recruited to receive one of our ChAdOx1 vaccines against a different pathogen.

4) The cut-off for the CHIKV E2 ELISA in Figure 3 should be better explained. It looks as though the cut-off was made just above the highest day 0 sample, yet there is significant variability in the day 0 samples.

We have added an explanatory paragraph on how the cut-off was calculated within the ELISA methods.

Briefly, the assay cut-off was determined from the analysis of all 24 pre-vaccinated (Day 0) samples of the trial volunteers and determined mathematically using the mean plus three standard deviations of the EU values. This value was defined as the threshold of the assay, from which detection was feasible.

5) It seems counter-intuitive to have lower ELISA seroconversion compared to PRNT50 seroconversion. Have the authors analyzed the data by normalizing to each individual subject's day 0 titer as a baseline?

Since the ELISA reported in this manuscript is against a single chikungunya envelope protein (E2), there is the possibility that we might be missing the detection of antibodies against other structural antigens. There is also the possibility that conformational epitopes might be underrepresented or lost in ELISA assays. Detection issues and bias between linear and conformational epitopes when bound to plastics (i.e. ELISA plate) is a well-known phenomenon.

Although antibodies against CHIKV E2 are considered a good indicator of humoral immunogenicity and protection (Fox et al. *Cell* 2015 163(5):1095-1107; Lum et al. *J. Immunol* 2013, 190(12):6295-6302; Kam et al. *EMBO Mol Med* 2012, Apr 4(4):330-343; Quiroz et al. *PLoS Pathog* 2019, 15(11): e1008061), the gold standard correlate of protection is PRNT. Our results support a significant correlation between the detected anti-E2 antibody titres and PRNT₅₀ (Fig.3 c and d).

It is common practice to report ELISA units when performing standardised ELISAs (Folegatti, et al. *The Lancet* 2020, 20(7):816-826; Ewer, et al. *Nat Med* 2021, 27(2):270-278; Barouch et al. *The Lancet* 2018, 392(10143):232-243). However, as per reviewer's suggestion, we have created a normalised graph to each participant whereby we plot fold change from the ELISA units detected at baseline (day 0). The issue is that we are finding it difficult to define a mathematical threshold for this representation. Would the reviewer consider a 3 fold-change cut-off acceptable? Please note that statistical outputs are the same when plotting by ELISA units and by fold-change.

6) In panels c and d, would the PRNT50 data be better suited on a log x-axis? We have changed the X-axis in panels c and d to a log10 scale.

7) Line 116- AsAm- the abbreviation has not been introduced.

The definition for this abbreviation has now been incorporated.

8) Line 115- “Slightly lower seroconversion rates of 96.7%, 100% and 83.3% were observed against YO111213 (AsAm) on days 14, 28, and 182.” I’m assuming the authors meant to include data for day 56 and not day 28? A 100% seroconversion rate is not lower than 100%! Also, 96.7% does not match the values in Table 3, which has 91.6%, 91.6% and 83.3 % seroconversion at days 14, 56 and 182, respectively, for the AsAm strain.

The authors acknowledge that the sentence was unclear. We have modified the sentence to: ‘PRNT₅₀ to YO111213, from the Asian-American Lineage (AsAm), demonstrated a 100% seroconversion rate on day 28 but slightly lower seroconversion rates on days 14, 56 and 182 (91.6%, 91.6% and 83.3%, respectively)’.

9) Line 122- Should “maximum PRNT50 GMT” be simply “maximum PRNT50”? It is either the maximum PRNT50 value chosen from all of the individual values within the group, or the GMT of the group, but these are exclusive data points. Please explain if otherwise.

The reviewer’s point is valid. We have now removed the word ‘GMT’.

10) Line 137- typo- Withing should be within

The typo has now been corrected.

General changes

1) Abstract

The length of the abstract has been reduced to 150 words as per journal requirements.

2) References

The list has been updated to include the references suggested by the reviewers. Formatting was adjusted as per journal requirements.

3) Statistics

Every statistical value was reviewed and formatted to comply with journal requirements (‘p=’ was replaced by ‘P=’). The statistics section, within methods, was extended to reflect each of the statistical tests used.

4) Display of axis in Figures

Some axis have been changed to achieve uniformity and to follow reviewer’s suggestions.

Reviewers' Comments:

Reviewer #1:

Remarks to the Author:

The issues have been addressed. One additional issue has come to my attention on re review. There would appear to be instances where the same data is presented in two different ways - in such instances (for instance, fig 3 a and b) - if correct, then the figure legend should state that b is the same data as a.

Andreas Suhrbier

Reviewer #2:

Remarks to the Author:

The authors have addressed my comments. Regarding the ELISA data, the new graph provided in which the results were normalized to each individual's baseline value is helpful and demonstrates the rise in antibody titers on an individual level. My concern is that in Figure 3a/b, the seropositivity cutoff seems arbitrarily high, as ELISA titers are consistently increasing following vaccination, but fall below the cutoff line for "seropositivity". Looking at the high dose group in Fig. 3b, for example, the data just does not match the conclusion that seropositivity is not achieved until 56 days post vaccination, nor does it match our general understanding of the kinetics of the antibody response- the antibody titers are clearly increasing following vaccination. While a conservative analysis of the data is generally preferred, in this case it seems almost too conservative. Perhaps the better question is why some individuals have such high ELISA background? Are the titers in these high background individuals also boosted by vaccination/ do they correspond with the highest PRNT titers? Nonetheless, I agree with the authors that the ELISA data is secondary in importance to the PRNT data, which does indicate no neutralization at day 0. I personally think the current Figure 3b would be better replaced by the new fold-change graph. At the very least, a comment about the variable ELISA background at day 0 could suffice.

THE JENNER INSTITUTE

Centre for Cellular and Molecular Physiology.
Wellcome Centre for Human Genetics Building.
Roosevelt Drive, Oxford OX3 7BN

Dr. Arturo Reyes-Sandoval
Associate Professor
tel: +44(0)1865 2 87811
arturo.reyes@ndm.ox.ac.uk

25th June 2021.

Manuscript # NCOMMS-21-05382B - Final Revision

“A single dose of ChAdOx1 Chik vaccine induces broadly neutralising antibodies against four chikungunya virus lineages in a phase 1 clinical trial”.

Editorial office

Dear reviewers,

Below you will find the point-by-point response to each of your final comments. We trust that the improvements made to our revised manuscript would render it suitable for publication in Nature Communications.

Reviewer #1

We thank the reviewer for the careful and considered critique of our revised manuscript.

- 1) *One additional issue has come to my attention on re review. There would appear to be instances where the same data is presented in two different ways - in such instances (for instance, fig 3 a and b) - if correct, then the figure legend should state that b is the same data as a.***

As requested, we have clarified within the figure legends if the datasets are the same but are presented and/or analysed in a different way.

Reviewer #2

We thank the reviewer for the careful and considered critique of our revised manuscript.

- 1) *Regarding the ELISA data, the new graph provided in which the results were normalized to each individual's baseline value is helpful and demonstrates the rise in antibody titers on an individual level. My concern is that in Figure 3a/b, the seropositivity cutoff seems arbitrarily high, as ELISA titers are consistently increasing following vaccination, but fall below the cutoff line for “seropositivity”. Looking at the high dose group in Fig. 3b, for example, the data just does not match the conclusion that seropositivity is not achieved until 56 days post vaccination, nor does it match our general understanding of the kinetics of the***

THE JENNER INSTITUTE

Centre for Cellular and Molecular Physiology.
Wellcome Centre for Human Genetics Building.
Roosevelt Drive, Oxford OX3 7BN

Dr. Arturo Reyes-Sandoval
Associate Professor
tel: +44(0)1865 2 87811
arturo.reyes@ndm.ox.ac.uk

antibody response- the antibody titers are clearly increasing following vaccination. While a conservative analysis of the data is generally preferred, in this case it seems almost too conservative. Perhaps the better question is why some individuals have such high ELISA background? Are the titers in these high background individuals also boosted by vaccination/ do they correspond with the highest PRNT titers? Nonetheless, I agree with the authors that the ELISA data is secondary in importance to the PRNT data, which does indicate no neutralization at day 0. I personally think the current Figure 3b would be better replaced by the new fold-change graph. At the very least, a comment about the variable ELISA background at day 0 could suffice.

We acknowledge the reviewer's comment on the observed background at baseline. We have validated that the samples were seronegative for CHIKV by using two commercially available ELISA kits. In line, we have included a new paragraph within the results section:

"It was observed that the calculated cut-off threshold (mean on day 0 + 3 SDEV), was influenced by 4 participants with a relatively high ELISA background at baseline. Therefore, we decided to further validate the seronegativity of these individuals with two commercially available ELISA kits. Both anti-chikungunya virus IgG ELISA kits, from Abcam and Euroimmune, confirmed that none of the participants that had high background in our in-house ELISA were seropositive for CHIKV at baseline (data not shown)."

As suggested, we have replaced panel 3b with a representation of the data in fold-change from baseline.